# Rotations improve the diversity of rhizosphere soil bacterial communities, enzyme activities and tomato yield

Cui Feng[1⊚], Zhengwei Yi[1⊚], Wei Qian[1], Huiying Liu[1], Xiaosan Jiang[2]*

1 Taizhou Institute of Agricultural Sciences, Jiangsu Academy of Agricultural Sciences, Taizhou, China,
2 College of Resources and Environmental Sciences, Nanjing Agricultural University, Nanjing, China

⊚ These authors contributed equally to this work.
* gis@njau.edu.cn

**Data Availability Statement:** All relevant data are within the manuscript and its Supporting information files.

## Abstract

The use of rotations is an effective strategy to control crop diseases and improve plant health. The soil bacterial communities in the rhizosphere are highly important for maintaining soil productivity. However, the composition and structure of soil bacterial communities in the rotations of vegetable crops remain unclear. In this study, we explored the bacterial diversity and community structure of the tomato rhizosphere, including enzyme activities, yield, and fruit quality, under three different cropping systems: tomato-tomato (*Solanum lycopersicum*) continuous cropping (TY1), eggplant (*Solanum melongena*)-tomato rotation (TY2) and arrowhead (*Sagittaria trifolia*)-tomato rotation (TY3). The composition and diversity of the rhizosphere bacterial communities differed significantly. The diversity was more in the TY2 and TY3 treatments than those in the TY1 treatment. *Chujaibacter* and *Rhodanobacter* were two predominant and unique strains detected only in TY1, while the relative abundances of *Curvibacter* and *Luteimonas* were the highest in TY2 and TY3, respectively. Moreover, *Lysobacter* was a relatively abundant type of biocontrol bacterium found only in the TY3 treatment, which could contribute to alleviating the obstacle of tomato continuous cropping. Compared with the TY1 treatment, the activities of catalase were significantly higher in the TY2 and TY3 treatments. In addition, compared with TY1, the TY2 and TY3 plots increased the following parameters: tomato yields by 24–46%, total soluble solids by 37-93%, total organic acid by 10-15.7% and soluble protein by 10-21%, while the content of nitrate was significantly reduced by 23%. Altogether, compared with the tomato monoculture, the rotations of tomato with eggplant and arrowhead shifted the rhizosphere bacterial communities and improved the yield and quality of the tomato. Moreover, a tomato rotation, particularly with arrowhead, was an effective way to alleviate the obstacles of continuous cropping.

## Introduction

Tomato (*Solanum lycopersicum* Mill.) is an important vegetable in solar greenhouses, and it is rich in lycopene, which is antibacterial and has shown potential to be against cancer [1]. In

**Funding:** This research was funded by the Excellent teaching team of "Qinglan Project" of Jiangsu Province (2020 No.10). The funders had no role in study design, data collection and analysis, decision to publish, or preparation of the manuscript.

**Competing interests:** The authors have declared that no competing interests exist.

recent years, greenhouse tomato planting has developed rapidly since it not only allows intensive production but also produces high-yield and quality vegetables without seasonal restrictions. However, the area of greenhouse tomato monoculture resulted in soil degradation and salinization [2], which lead to an imbalance of the soil micro-ecosystem, and enrichment of harmful microorganisms [3, 4]. Soil-borne pathogens, particularly *Fusarium* species, cause serious damage to tomato plants, including *F. oxysporum* f. sp. *lycopersici*, the causal agent of Fusarium wilt (FW), and *F. oxysporum* f. sp. *radicis-lycopersici*, the causal agent of Fusarium crown and root rot (FCRR) [5, 6]. Therefore, it is urgent to adopt appropriate soil management measures to alleviate these continuous cropping obstacles.

Crop rotation has been identified as an alternative strategy to maintain soil quality and vegetable productivity [7]. Previous studies have shown that reasonable crop rotation can improve the soil microecological structure increase the number of soil bacteria, alter the community structure [8, 9], enhance soil enzyme activity, and promote crop growth, which can effectively reduce continuous cropping obstacles [10, 11]. To improve soil microbial diversity by rotation, it may require the use of specific crop combinations, which are expected to have a greater impact on soil microbial diversity [12]. The cultivation of maize (*Zea mays* L.) in rotation with tomato reduced the populations of nematodes [13], while the cultivation of banana (*Musa* spp.) in rotation with pineapple (*Ananas comosus*) reduced the amount of FW pathogens in the soil fungal community [14]. These examples highlight the importance of crop rotations in the suppression of soilborne diseases. One of the inhibitory mechanisms is related to changes in the abundances of beneficial and pathogenic soil bacteria [15].

Soil microbes influence crop productivity, nutrient uptake and agricultural sustainability and play important roles in nutrient cycling [16, 17]. Certain species of soil microbes can protect plants from soilborne pathogens and improve plant growth by acting as biocontrol agents (BCAs) [18]. For example, the fungal genus *Mortierella* appears to enhance the suppression of FW disease in vanilla (*Vanilla planifolia*) [19]. Members of the bacterial genera *Pseudomonas* and *Bacillus* have been verified as two representatives for effective BCAs against different fungal pathogens, including the tomato FW pathogen [20]. In recent years, the interaction between rhizosphere organisms has become the subject of increasing amounts of researches [21]. The rotation of tomato with xerophytes or rice have been assessed by many studies [22–24], but there are few studies on rotation with aquatic vegetables. Arrowhead (*Sagittaria sagittifolia*) is an exceptional aquatic vegetable in the Lixiahe region, which is rich in starch, protein, vitamins and other nutrients [25]. Arrowhead is a water-loving plant, so it is a good choice for crop rotation [26]. However, there are few studies on the cultivation of *S. sagittifolia* and tomato, particularly on the effects of crop rotation on soil microbial and physicochemical properties. In addition, the effect of this type of crop rotation on bacteria is unclear.

Soil enzymes increase the rate at which plant residues decompose and release their available nutrients [27, 28]. Soil enzyme activity is the cumulative effect of long-term microbial activity and the activity of viable populations at sampling [29]. It plays an important role in the soil ecosystem, including the cycling of carbon (C), nitrogen (N), phosphorus (P) and plant nutrient transformations, which provide information on the ability of soils to perform biogeochemical reactions [30]. Furthermore, many studies have illustrated that human activities, such as field management measures, can significantly affect the activities of soil enzymes [31]. Different farming methods may directly or indirectly affect soil enzyme activities, including the soil physicochemical properties, vegetation types, plant roots, and soil microorganisms [32]. Therefore, the soil enzyme activities affected by rotation can be used to describe the changes in soil quality owing to continuous cropping to better understand the function and quality of soil ecosystems.

In this study, we evaluated the effects of different vegetable rotations (monoculture of tomato and rotations of eggplant [*Solanum melongena*]-tomato and arrowhead [*Sagittaria sagittifolia*]-tomato) on soil bacterial communities, and evaluated the most effective tomato rotation system. 16S rRNA gene was used to estimate the microbial community structures. The soil enzyme activity, tomato yield and fruit quality were measured simultaneously. We aimed to (i) examine the response of soil bacterial communities to varied tomato rotations and identify significant differences in the composition of taxa, (ii) evaluate the variation in soil enzyme activity under these three rotations, and (iii) compare the changes in tomato yield and quality. This study is the first to analyze the effects of arrowhead-tomato rotation on tomato growth and rhizosphere soil environment, and will provide insights into the effectiveness of tomato rotation systems in alleviating the productivity obstacles caused by tomato monoculture.

## Materials and methods

### Study site description

A plastic greenhouse experiment was conducted at an experimental plot of the Taizhou Academy of Agricultural Sciences, Taizhou, Jiangsu Province, China (32°54′N, 11°99′N). The greenhouse monoculture of tomato had been used in this plot for more than two years before this experiment was established. The area is a subtropical monsoon climate, and the annual mean temperature, annual mean accumulated temperature and sunshine duration were 14.5°C, 5,365.6°C and 2,205.9 h, respectively. The annual mean precipitation was 991.7 mm, and there were 117 annual mean rainy days. The USDA classification system identified the soil as loamy. The surface soil (0–20 cm) in the greenhouse had a pH of 7.8, 25.8 $g \cdot kg^{-1}$ organic matter, 1.68 $g \cdot kg^{-1}$ total N (TN), 16.5 $mg \cdot kg^{-1}$ available P (AP), and 211 $mg \cdot kg^{-1}$ available K (AK) before the experiment was initiated.

### Experimental design

The field experiment was conducted from July 2018 to July 2020, and two crops of tomatoes were planted from July 2018 to July 2019. The continuous (round) planting period of the experimental vegetables was from August 2019 to January 2020, and the growing period of experimental tomatoes was from March 2020 to July 2020. Three vegetable rotations, consisting of monoculture tomato (Japanese Fenwang, TY1), eggplant (Suqi 3)-tomato rotation (TY2), and arrowhead-tomato rotation (TY3), were arranged in a randomized block design with three replicates. Each plot within the plastic greenhouse was 12 m × 1.2 m. There were 48 tomato plants on each plot, and the distance between the two plants was 50 cm. A 0.5 m wide buffer row separated the adjacent plots. In addition, the buffer line was wrapped with black plastic film to avoid mutual interference between the plots. The field management options, including transplantation, irrigation and the use of pesticides, were consistent with the local cultivation standards. Among them, arrowhead was grown in moist culture.

### Soil sampling

Five rhizosphere soil subsamples were randomly collected from each replicate plot and mixed as a composite sample for analysis at the tomato harvest stage. The collection of rhizosphere soil involved firstly removing approximately 3 cm of the surface soil. The entire tomato root was taken out, and the rhizosphere soil was separated from the bulk soil by gently shaking. All the soil samples were screened via a 2 mm sieve and thoroughly homogenized after all the visible roots and impurities had been removed. One portion of the homogenized fresh soil sample

was stored at −70˚C for DNA extraction, and the other one was used immediately to determine the soil enzyme activity.

## Measurements and methods

### Extraction and sequence of soil microbial DNA

The soil DNA was extracted using a Fast DNA Spin Kit for Soil (MP Biomedicals, Cleveland, OH) according to the manufacturer's instructions. The quality of DNA was detected by 0.8% agarose gel electrophoresis and quantitatively analyzed by NanoDrop ND-1000 (Thermo Fisher Scientific, Waltham, MA, USA). The barcoded primers 16S5/16S2R (5′- GGAAG TAAAAGTCGTAACAAGG—3′/5′- CTGCGTTCTTCATCGATGC -3′) were used to amplify 2 μL of DNA template, 0.25 μL of Q5 DNA polymerase, and 8.75 μL of ddH2O. The cycling program was 98˚C for 2 min; 27 cycles of 98˚C for 15 s, 55˚C for 30 s, 72˚C for 30 s; and 72˚C for 5 min. The PCR amplification products were detected by 2% agarose gel electrophoresis, and the target fragments were recovered by cutting the gel and using a gel recovery kit from Axygen (Axygen Biosciences, Union City, CA, USA). The amplicon libraries were pooled and sequenced using a $2 \times 250$ base paired Illumina MiSeq platform (Illumina, Inc, San Diego, CA, USA) and then stored at 10˚C.

### Soil enzyme assays

The activities of urease, catalase, sucrase and alkaline phosphatase were determined as described by Hu et al. (2014) [33] with some modifications. The urease activity was analyzed using the phenol-sodium hypochlorite colorimetric method. In brief, 10 mL of 10% urea solution and 20 mL of citrate buffer (pH 6.7) were added to a 50 mL conical flask that contained 5 g of soil. After 24 h of incubation at 37˚C, the suspension was filtered, and then 0.5 mL of the filtrate was added to a 50 mL flask that contained 20 mL of distilled water and 4 mL of 1.5 M sodium phenol solution. Next, 3 mL of 0.9% sodium hypochlorite solution was added to the flask and mixed. The volume was adjusted to 50 mL with distilled water, and the absorbance was monitored at 578 nm. The catalase activity was measured using the potassium permanganate ($KMnOM_4$) titration method. Briefly, 5 mL of 0.5% $H_2O_2$ was added to 1 g of soil for 20 minutes, and the residual $H_2O_2$ in the soil was titrated with 0.01 M KMnO4. The $H_2O_2$ consumed was expressed as catalase activity. The 3,5-dinitrosalicylic acid colorimetric method was used to analyze the soil sucrase activity. A volume of 0.2 mL of toluene and 5 mL of N-acetyl-muramic acid (MUB) were added to a 50 mL Erlenmeyer flask that contained 3 g of soil and they were shaken for a few seconds to mix the sample. Next, 5 M of 10% sucrose was added and shaken again. The flask was sealed, and the soil sample was incubated at 37˚C for 24 h. A volume of 1 mL of filtrate, 2 mL of 2 M NaOH, 2 mL of the color reagent and 5 mL of deionized water were added to a 50 mL test tube and flushed with $N_2$ for 10 min. The tubes were all placed in a boiling water bath for 5 min with inverted 50 mL beakers on their tops and then cooled to room temperature. A spectrophotometer was used to determine the color intensity at a wavelength of 540 nm. The soil phosphatase activity was determined using the disodium phenyl phosphate method. In short, 5 g of soil sample was added to a 250 mL flask, and 2 mL of toluene was used to inhibit the growth of microorganisms. After standing for 15 min, 20 mL of 0.5% (w/v) disodium phenyl phosphate prepared in acetic acid buffer (pH 5) was added to the flask and then incubated at 37˚C for 24 h. Subsequently, after adding 100 mL of 0.3% $Al_2(SO_4)_3$ to the flask, the soil sample was filtered, and 3 mL of the filtrate was added to a 50 mL flask. A volume of 5 mL of borate buffer (pH 9.4) and four drops of indicator were added. The volume was adjusted to 50 mL, the absorbance at 660 nm was determined.

## Determination of vegetable quality

The content of VC was determined by 2, 6-dichloroindigophenol titration [34]. The content of TSS was determined by refractometry [35]. The content of titratable acid was determined by acid-base titration [36]. The nitrate content was determined by UV spectrophotometry [37]. The soluble protein content was determined with Coomassie brilliant blue using bovine serum albumin as the standard [38]. The soluble total sugar content was determined by the anthrone method [38].

## Data analysis

PRINSEQ, PANDAseq, USEARCH, and self-developed Perl were used to analyze the microbial diversity of sequencing results. The data of tomato yield and quality were processed using Microsoft Excel 2010 (Redmond, WA, USA). SPSS 18.0 (SPSS, Inc., Chicago, IL, USA) was used for statistical analyses. Duncan's new multiple range test was used to test the significance of difference. $P < 0.05$ was considered as the significant difference.

## Soil bacterial community diversity among tomato planting patterns

All the samples were rarefied to the same number of reads as the sample with the lowest reads, which was 168,816. The bacterial sequences were clustered at 97% similarity and annotated in the RDP database to obtain 2,673, 3,074, and 3,266 bacterial operational taxonomic units (OTU) for TY1, TY2, and TY3, respectively (Table 1). The coverage rates of the 16S rRNA sequencing in the three soil samples were 0.9992, 0.9995, and 0.9996, respectively, indicating that the sequencing depth had basically covered all the species in the sample.

The diversity analysis indicated that TY1 (802) had the lowest Chao1 index, while TY3 (1,065) had the highest Chao1 index, and the Chao1 index of TY2 (996) was between those of TY1 and TY3. From TY1 to TY3, the Shannon index gradually increased (5.1, 6.7, and 7.1, respectively). It is apparent that the diversity and richness of tomato soil bacteria after rotation were higher than those of continuous cropping, and the diversity and richness of soil bacteria were the highest under the mode of arrowhead-tomato rotation. The bacterial diversity and richness decreased under continuous cropping, and the rotation could increase the microbial species in soil.

The Venn diagram intuitively showed the number of common and unique OTUs of different samples (Fig 1). TY1, TY2, and TY3 contained 2,673, 3,074 and 3,266 OTUs, respectively, with 2,125 OTUs shared by the three soil samples. Thus, more than 79% of the OTUs were the same in each soil. The number of specific bacterial OTUs was 142, 221, and 465, respectively. TY3 had the highest number of specific bacterial OTUs, which was 3.27-fold higher than that of TY1, indicating that more specific bacterial groups existed in the arrowhead-tomato rotation.

**Table 1. Soil bacterial diversity indices of different planting patterns.**

| Treatment | Sequence numbers | OTU numbers | Chao1 index | Shannon index | Simpson index | goods coverage |
|-----------|-----------------|-------------|-------------|---------------|---------------|----------------|
| TY1 1 | 168816 | 2,673 | 802 | 5.1 | 0.92 | 0.9992 |
| TY2 2 | 168816 | 3,074 | 996 | 6.7 | 0.97 | 0.9995 |
| TY3 3 | 168816 | 3,266 | 1065 | 7.1 | 0.98 | 0.9996 |

[1] Tomato continuous cropping;

[2] Eggplant-tomato rotation;

[3] Arrowhead-tomato rotation.

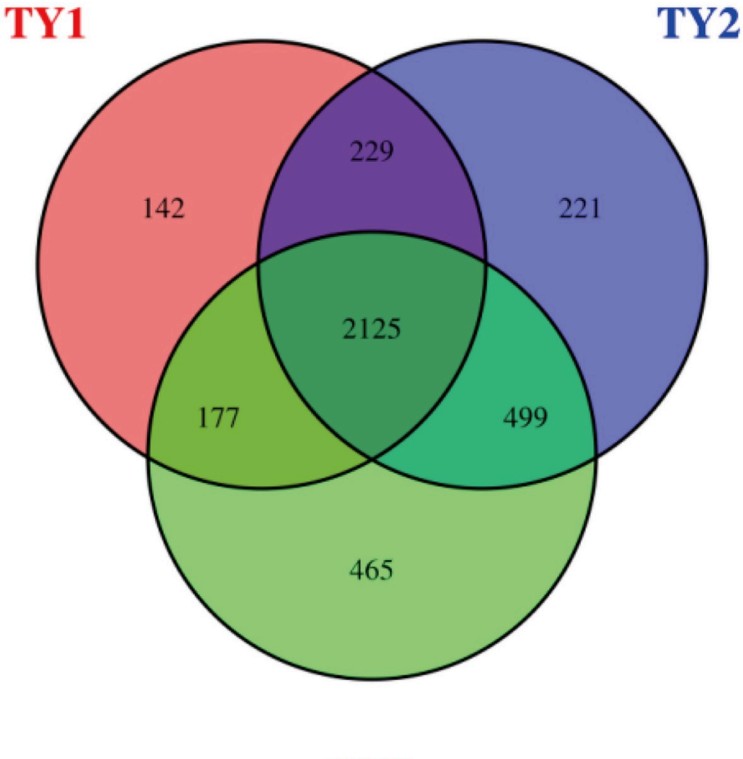

**Fig 1. Venn diagram showing the distribution of OTUs in different soil samples.** TY1, Tomato continuous cropping; TY2, Eggplant-tomato rotation; TY3, Arrowhead-tomato rotation.

## Comparison of the soil bacterial in different soil samples at the level of genera

The genera composition of soil bacteria showed that there were at least 769 bacterial genera in all the soil bacterial communities, and the first 21 genera based on relative abundance were selected for comparison. In the identified bacterial community, *Chujaibacter* and *Rhodanobacter* were the two predominant and unique strains detected only in TY1, while no such bacteria were found in the other two crop rotation patterns (Fig 2). Some studies have shown that species of *Rhodanobacter* positively correlated with the incidence of FW [39]. An analysis based on OTUs within the species abundance information was visualized at the genus level with the five most abundant genera. The abundance of soil bacterial genera was TY1>TY2>TY3. TY1 had the highest relative abundance of *Chujaibacter*, while TY2 had the highest abundance of *Curvibacter* (Fig 3). *Luteimonas* was the most abundant in TY3, and *Lysobacter* was only found in TY3. These results indicated that there were differences in the species, diversity, and richness of soil bacteria among different cropping patterns, and the rotation helped to enhance the numbers of beneficial soil bacteria.

## Distribution curves of soil microbial abundance among different planting modes

The richness and evenness of species and the TY3 treatment curve is relatively flat (Fig 4), indicating that the soil bacterial composition is the highest uniformity under the irrigated and upland

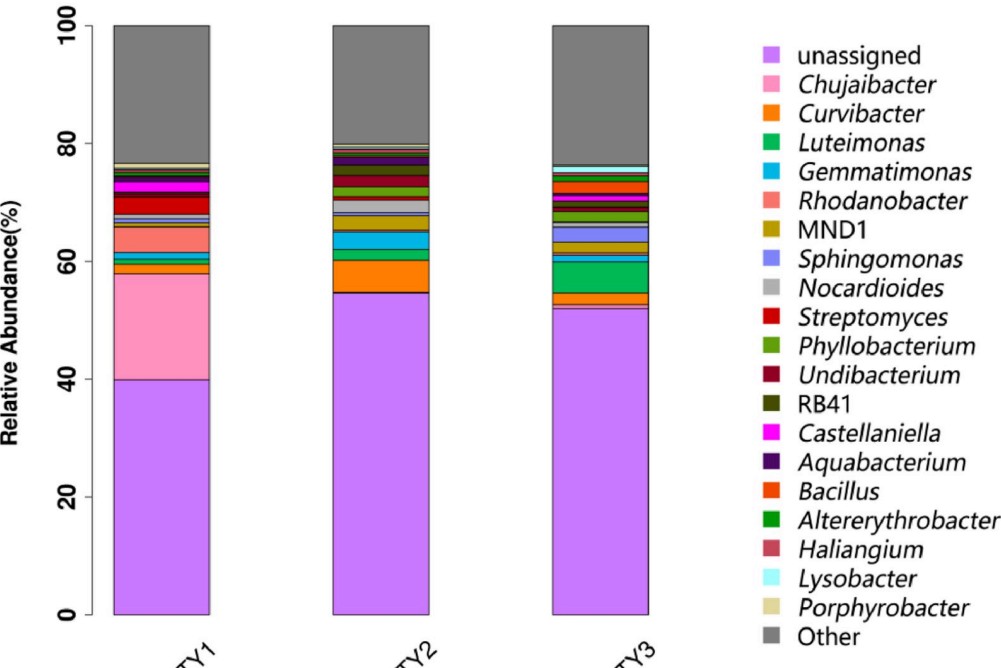

**Fig 2. Comparison of bacterial classification at the genus level in soil of different planting patterns.** TY1, Tomato continuous cropping; TY2, Eggplant-tomato rotation; TY3, Arrowhead-tomato rotation.

rotation mode. The curve width of TY3 and TY2 treatment was significantly wider than TY1, indicating that the soil bacterial species diversity and richness of TY3 and TY2 were higher.

## Effects of different planting patterns on soil enzyme activities

The activity of urease differed significantly among the three planting patterns, and the amount of activity was as follows: TY1>TY2 >TY3 (Fig 5). There were significant differences in catalase activity among the three cropping modes (P<0.05). TY3 had the highest level of catalase activity. In general, soil that was more fertile had higher amounts of sucrase activity. The sucrase activities of the three samples were as follows: TY3>TY1>TY2. Phosphatase can be divided into three types: acid phosphatase, neutral phosphatase and alkaline phosphatase. The soil samples collected in this experiment were all alkaline soil samples with pH values > 7, so only alkaline phosphatase activity was measured. The results of the three soil samples show that the amount of alkaline phosphatase activity in TY1, TY2 and TY3 differed significantly (P<0.05). TY2 had the highest activity of alkaline phosphatase, while there was no significant difference with TY1 (P<0.05).

## Effects of different planting patterns on the yield and quality of tomato

There were significant differences in tomato yield among the three cultivation patterns (Fig 6). Compared with continuous cropping, the eggplant-tomato and arrowhead-tomato rotations increased the yield of tomato by 28.6% and 16.3%, respectively, indicating that rotating the crops helped to increase the yields of tomato.

There were some differences in tomato quality under the three different cultivation patterns (Fig 7). In general, the contents of TSS, VC, soluble proteins and soluble total sugars in the TY2 tomatoes were all higher than those in TY1, indicating that crop rotation helped to

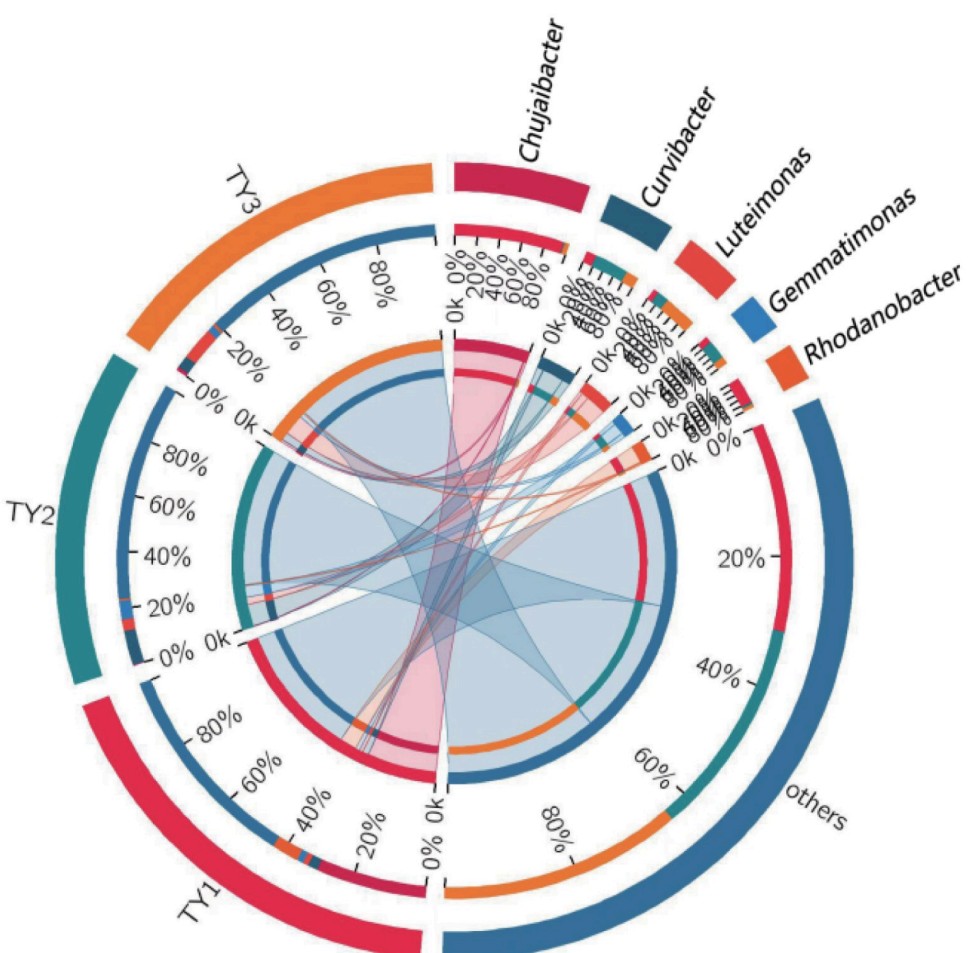

**Fig 3. Profiling circles of the top five species in the different sample genera classification levels.** TY1, Tomato continuous cropping; TY2, Eggplant-tomato rotation; TY3, Arrowhead-tomato rotation. In the graph of collinearity between samples and species, the left semicircle represents the species abundance composition of samples, and the right semicircle represents the distribution proportion of species in different samples under this clustering level.

improve fruit quality. The nitrate content of tomato under the different cultivation patterns differed, and the difference between an irrigation-upland rotation and continuous cropping was highly significant (P<0.05), and TY3 was the lowest, indicating that an irrigation-upland rotation could reduce the nitrate content of tomato fruit.

## Discussion

Bacteria are the most widely distributed and abundant components of soil microorganisms [12], accounting for 70%-90% of the total number of soil microorganisms. This has resulted in their use as one of the most sensitive indicators for changes in soil nutrients [39]. This study used high-throughput sequencing technology to compare the differences in the community structure and abundance of soil bacterial species under different tomato planting patterns. Our results showed that the number of bacterial OTUs in the TY1, TY2 and TY3 samples were 2,673, 3,074 and 3,266, respectively. The Chao1 and Shannon indices ranked TY3>TY2>TY1 (Table 1). This indicated that the soil bacterial community structure and abundance differed somewhat between the rotation patterns and continuous cropping [40]. Crop rotation can

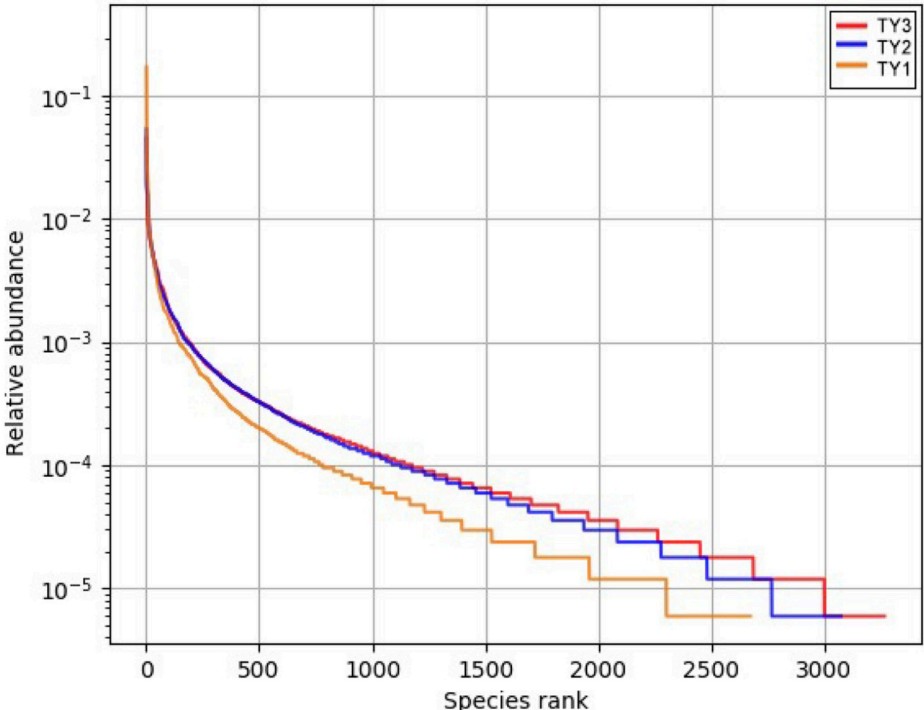

**Fig 4. Curve of the abundance distribution for three samples.** The horizontal axis shows the relative abundance of OTU in descending order, while the vertical axis shows the proportion of relative abundance of OTU. TY1, Tomato continuous cropping; TY2, Eggplant-tomato rotation; TY3, Arrowhead-tomato rotation.

improve the microbial diversity and increase the abundance of soil bacteria [41], especially under the irrigation-drought rotation mode. As the aquatic vegetable stay flooded for most of the cropping season, the soils are unique in their microbial ecology, as diversity of the microorganisms inhabiting these soils is crucial for assuring nutrient cycling, soil fertility, among other vital functions [42, 43]. This is also consistent with the results of Liu et al. (2021) [44] who found that the abundance of bacteria in tomato continuous cropping and rotation soil differed substantially. Our results showed that the three soil samples had different dominant genera, and the rotation system had more beneficial and diverse bacterial species. *Rhodanobacter* was the dominant genus in TY1 and is a member of the family Xanthomonadaceae; it positively correlated with the incidence of FW [22]. *Aquabacterium*, *Luteimonas* and *Gemmatimonas* were found in tomato continuous cropping, and some metabolites of these bacteria are antimicrobial [45]. They were capable of controlling Fusarium wilt of tomato [46]. Some species of *Gemmatimonas* were proven to prevent disease, promote growth, remediate heavy metal pollution, and decompose toxic gases [44]. *Lysobacter* can resist adverse environments and prevent soil diseases, which is a unique strain of TY3 [47]. Therefore, tomato continuous cropping can lead to the accumulation of harmful bacteria in the soil, while crop rotations can change the bacterial structure in the soil, which help to increase the numbers of biocontrol bacteria and other types of beneficial bacterial species. This is consistent with the results of previous studies [48, 49]. Lee H [50] found that the incidence of bacterial wilt, blight and root-knot nematode disease in the tomato rotation fields were 12%, 16% and 8% lower than those in the conventional fields, respectively. The distribution curves of the soil microbial abundance showed that the diversity and evenness of the soil bacterial species and the abundance of each species under rotation were higher than those under continuous cropping, and the evenness of

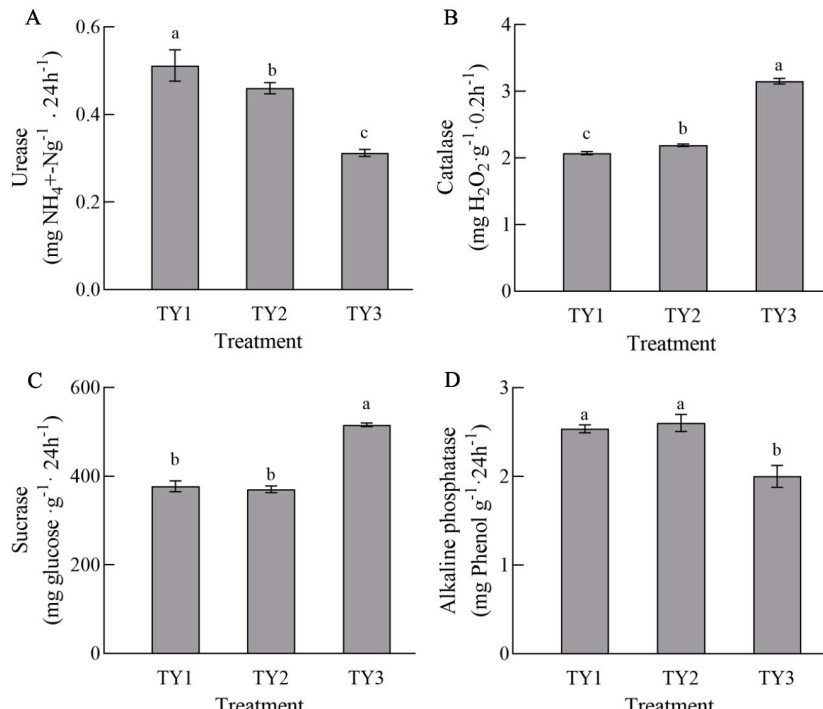

**Fig 5. Effects of different planting patterns on the enzyme activities of soil.** Different lowercase letters in the same panel indicate significant differences at the 0.05 level. In each panel, bar charts labelled by different letters indicate significant differences according to a one-way analysis of variance (ANOVA) ($P<0.05$) and Duncan's Multiple Range Test (DMRT) ($P < 0.05$). TY1, Tomato continuous cropping; TY2, Eggplant-tomato rotation; TY3, Arrowhead-tomato rotation.

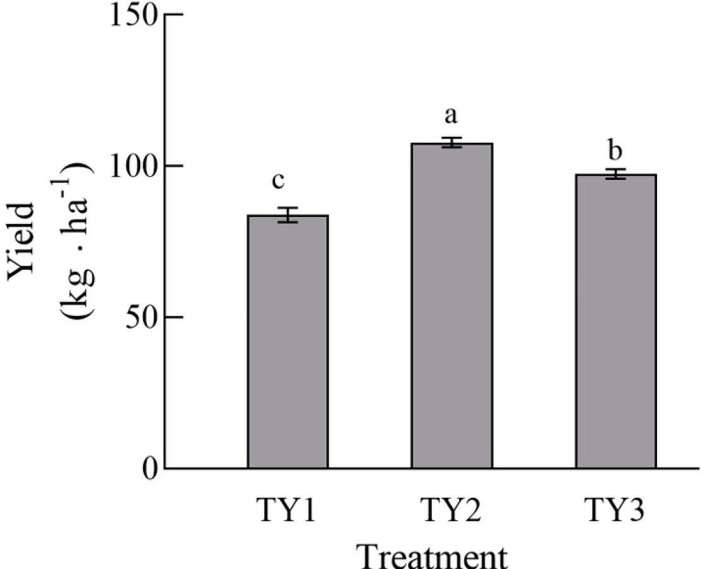

**Fig 6. Comparison of tomato yields between different planting patterns.** Different lowercase letters in the same panel indicate significant differences at the 0.05 level. In each planting method group, bar charts labelled by different letters are significantly different according to a one-way analysis of variance (ANOVA) ($P<0.05$) and Duncan's Multiple Range Test (DMRT) ($P < 0.05$). TY1, Tomato continuous cropping; TY2, Eggplant-tomato rotation; TY3, Arrowhead-tomato rotation.

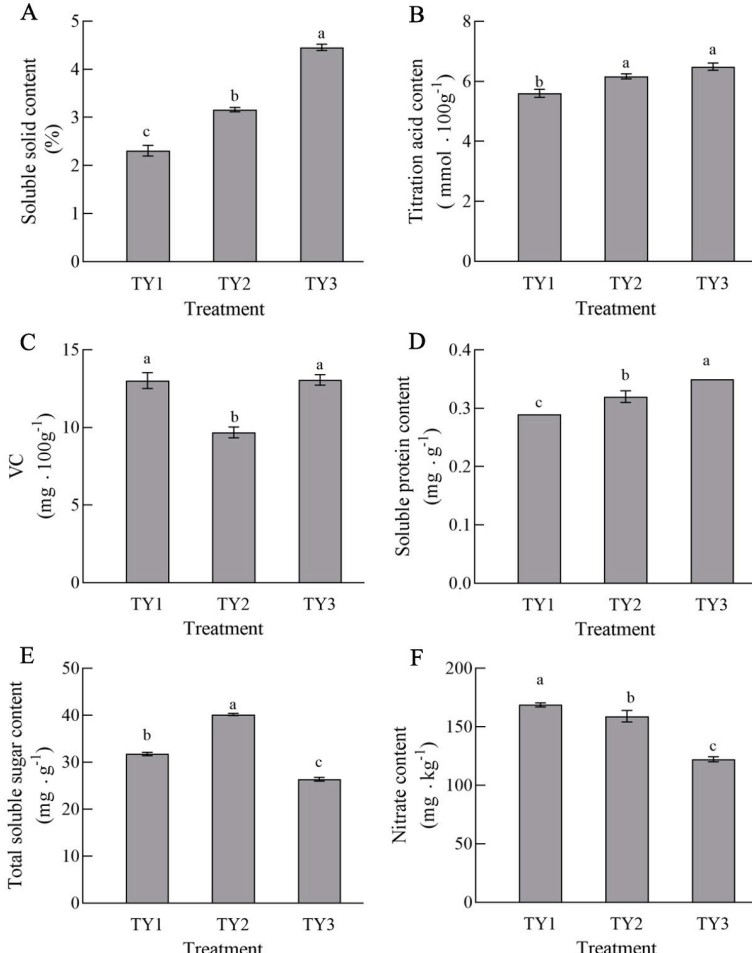

**Fig 7. Comparison of tomato quality among different planting patterns.** (a) soluble solids, (b) titration acid, (c) VC content, (d) soluble protein, (e) total soluble sugar, (f) nitrate content. Different lowercase letters in the same column indicate significant differences at 0.05 level. For each planting method group, bar charts labelled by different letters are significantly different according to one-way analysis of variance (ANOVA) ($P<0.05$) and Duncan's Multiple Range Test (DMRT) ($P< 0.05$). TY1, Tomato continuous cropping; TY2, Eggplant-tomato rotation; TY3, Arrowhead-tomato rotation.

soil bacterial composition was higher under the arrowhead-tomato rotation, which was much more likely to contribute to the control of microbial diseases. This can be explained by the increase in soil microorganisms of some types of beneficial bacteria that thrive under rotation cultivation and are beneficial for the crop growth [20, 51].

Soil enzymes are the most active compounds in the soil and important indicators that reflect biochemical processes in the soil [52], so they play an important role in the material cycle and energy transformation of the soil ecosystem. Studies have shown that years of facility planting and cultivation methods have important effects on soil microorganisms and enzyme activities [43, 53]. Lalith M. Rankoth et al. (2019) [54] found that the activities of catalase, urease and sucrase in the rotation soil were significantly higher than those in continuous cropping soil, which were similar to the results of this study. The changes in enzyme activity could be caused by continuously cropping tomato, which resulted in root secretions that altered the quantity of soil organic matter and caused changes to the microflora. Urease is the main source of available nitrogen in soil, which is conducive to the conversion of organic nitrogen to

ammonium nitrogen for plant growth and metabolism [55], and the activity of urease is an important indicator of the presence of nitrogen fertilizer. Phosphatase can hydrolyze soil organophosphorus compounds and release corresponding alcohols and inorganic phosphorus, and its activity is also one of the important indicators of soil fertility (particularly phosphate fertilizer) [39]. Our study revealed that the urease activity was the highest in TY1, and the phosphatase contents of TY1 and TY2 were similar. This is probably due to that both tomato and eggplant are solanaceous, and the short continuous cropping time can not distinguish between them. Catalase can destroy the hydrogen peroxide generated by biochemical reactions in the soil and reduce its deleterious effects on plants [55]. TY3 had the highest level of catalase activity, which could explain the greater diversity of microorganisms in the crop rotation and is consistent with the findings of previous research [56]. The strength of sucrase activity reflects the degree of soil maturation and fertility, and plays an important role in increasing the amount of nutrients in the soil [54]. The content of sucrase in TY3 was the highest. These factors indicate that a rotation of arrowhead and tomato could increase the soil fertility. However, some enzyme activities are not apparent, because soil quality is strongly associated with soil enzymes [57]. Further study will be needed.

The yield of tomato is an important index that reflects its economic value, and this study showed that crop rotation helped to improve the yields of tomato. Compared with continuous cropping, the yield of tomato increased by 28.6% and 16.3% in the eggplant-tomato and arrowhead-tomato rotations, respectively. Crude protein, VC and reducing sugar contents are important indicators that reflect the nutritional quality of vegetables, which affect their commodity value. Our study showed that there were some differences in tomato quality under three different cultivation patterns. In general, the contents of TSS, total organic acids, VC and soluble proteins in tomato under rotation were higher than those under continuous cropping. The nitrate contents of the eggplant-tomato and arrowhead-tomato rotations were 5.8% and 27.5% lower than that of continuous cropping, respectively. Nitrate is an important quality indicator of vegetables [58]. The activity of catalase was the lowest in TY1, which could be related to the formation of tomato continuous cropping obstacles. The activity of catalase was the highest in TY3, which indicated that the ability of an irrigation-drought rotation mode was stronger and better to be able to cleanse the soil, and it was one of the better cultivation methods to avoid the continuous cropping obstacles of tomato. Sucrase was the most active in TY3, and alkaline phosphatase was the most active in TY2, indicating that rotations are better in maintaining soil fertility [39, 54]. However, urease and phosphatase were the least active in TY3, indicating that crop rotation could affect the soil N and P cycles and promote the utilization of nutrients. In general, the activities of soil enzymes in the three soil samples did not show a strong trend, which could be owing to the short continuous cropping period of tomato in this study. Vegetables are the most important source of nitrate exposure in the human diet [59]. Excessive consumption of nitrate can cause human cyanosis and even induce cancers of the digestive system [58]. Sadeghi E et al. (2015) [59] showed that in non-organic planting systems, vegetables with a higher content of nitrate have a lower content of VC. Jacoby R et al. (2017) [60] found that improving the soil microbial community abundance and diversity can enhance disease resistance in the soil and increase the amounts of nutrients, which is consistent with the results of our study. Therefore, crop rotation can improve the yield and quality of tomato and reduce the nitrate content.

## Conclusion

This study evaluated the bacterial diversity and community structure of the tomato rhizosphere, including enzyme activities, yield, and fruit quality, under three different cropping

systems, and provided first-time insights in the effect of tomato and aquatic vegetable rotation on alleviating continuous cropping obstacle. Compared with tomato continuous cropping, the tomato rotation patterns significantly improved the abundance of beneficial bacteria. Moreover, the rotations increased the activities of soil enzymes, such as catalase, sucrase, and alkaline phosphatase. The rotations also improved the quality of fruit with an increase in the content of total soluble solids, soluble proteins, and VC, while it decreased the content of nitrate nitrogen. Thereby, the tomato rotation patterns improved the yield of tomato. It is worth noting that in the two types of rotation, arrowhead-tomato rotation was more effective at relieving the obstacles of continuous cropping. This study was the first to verify the effects of crop rotation of arrowhead and tomato on soil microbial and physicochemical properties. In practical production, the arrowhead-tomato rotation is recommended to promote crop growth.

## Supporting information

**S1 File. Microbial diversity in 3 soils (16s).**
(DOC)

**S2 File. Soil enzyme activities, tomato yields and quality.**
(XLSX)

## Author Contributions

**Formal analysis:** Cui Feng, Zhengwei Yi.

**Investigation:** Wei Qian, Huiying Liu.

**Methodology:** Cui Feng.

**Writing – original draft:** Cui Feng.

**Writing – review & editing:** Cui Feng, Xiaosan Jiang.

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
