## [Decision Letter · Decision Letter 0]

18 Aug 2022

PONE-D-22-17572Rotations improve the diversity of rhizosphere soil bacterial communities, enzyme activities and tomato yieldPLOS ONE

Dear Dr. Jiang,

Thank you for submitting your manuscript to PLOS ONE. After careful consideration, we feel that it has merit but does not fully meet PLOS ONE’s publication criteria as it currently stands. Therefore, we invite you to submit a revised version of the manuscript that addresses the points raised during the review process.

We look forward to receiving your revised manuscript.

Kind regards,

Raffaella Balestrini

Academic Editor

PLOS ONE

Journal Requirements:

Reviewers' comments:

Reviewer's Responses to Questions

**Comments to the Author**

1. Is the manuscript technically sound, and do the data support the conclusions?

Reviewer #1: Yes

Reviewer #2: Yes

2. Has the statistical analysis been performed appropriately and rigorously? 

Reviewer #1: I Don't Know

Reviewer #2: Yes

3. Have the authors made all data underlying the findings in their manuscript fully available?

Reviewer #1: Yes

Reviewer #2: Yes

4. Is the manuscript presented in an intelligible fashion and written in standard English?

Reviewer #1: Yes

Reviewer #2: Yes

5. Review Comments to the Author

Reviewer #1: The paper is interesting but there are few points that should be revised before publication. The authors should indicate which statistic test they have used to analyze the results (indicate p value also). In general the paper is written in a good English, but more references could be added to support the text. In particular, references in English would add value to the paper. There are ten references in Chinese, please find and include other references in English to support your sentences.

In the discussion section, you should not repeat the results without further discuss them. I think you should revise the discussion section trying not to repeat onlythe results (for example, have a look at lines 311-315)

Keywords: I suggest to use keywords that are not included in the article title. In this way it is probable that more people are reading your paper.

Line 31: “Lysobacterium” is rarely used in the literature. In order to avoid misunderstandings I suggest to replace it with the more used and known “Lysobacter”.

Line 74: please add references about tomato-rice and tomato-xerophyte crop rotation.

Lines 75-77: add references to support this sentence.

Lines 158-159 Specify the concentration of the sodium phenol solution and of sodium hypochlorite solution used.

Figure 4: if possible, please insert a bigger legend to the figure. The current legend is too small and difficult to read.

Lines 258-259: “TY3 had the highest level of activity.” What are you referring to in this sentence? Are you referring to catalase or sucrase? Please specify.

Lines 260-261: “[…] indicating that the arrowhead-tomato rotation could increase soil fertility.” I think that this sentence should be part of the discussion session. Moreover, you should include more data supporting this sentence to be able to arrive at this conclusion.

Lines 267-269: These sentences should be part of the discussion section.

Table 2: Please specify what are the numbers in the table and if they have a unit of measurement. In addition, I suggest to represent the table graphically. This would help the reader to understand quickly the results. Please, specify also the statistic test you used to analyze the data.

Figure 5 and Figure 6: Please specify which statistic test you use to analyze the data.

Figure 6: What is the difference between (e) and (f) ? They are both “Total soluble sugar content”.

Line 325: replace “benefit” with “beneficial”.

Lines 341 - 345: Can you find references to support these sentences that are not in Chinese language?

Lines 356 – 357: “Nitrate is an important quality indicator of vegetables.” Please include references to support this sentence.

Lines 361 -363: This sentence assumes that sucrase is an indicator of soil fertility. Please include references that support your assumption (possibly in English).

Lines 367 -368: I do not understand how this sentence correlates with the rest of the text. Please rephrase and add a reference in English.

Reviewer #2: Rotations improve the diversity of rhizosphere soil bacterial communities, enzyme activities and tomato yield

The agronomic crop rotation technique involving alternating different agricultural species on the same plot of land to rebalance the cultivated soil's biological, chemical and physical properties is crucial, especially with the increase in soil marginality.

The work presented concerns the experimentation of two types of plant species cultivated in alternation with tomatoes. The authors focused their activity on evaluating variations in microbial communities, soil enzyme levels and tomato yield in monoculture compared to rotation with eggplant and arrowhead.

General comments:

The manuscript is well structured. Both materials and methods and the results/discussion are detailed and precise.

Specific comments:

ABSTRACT

Lines 24-26: …" tomato-tomato continuous cropping (TY1), eggplant-tomato (S. melongena) rotation (TY2) and arrowhead-tomato (Sagittaria trifolia-Solanum lycopersicum) rotation (TY3)." Please standardize the descriptions. For example: tomato-tomato (Solanum lycopersicum) continuous cropping (TY1), eggplant (Solanum melongena) -tomato rotation (TY2) and arrowhead (Sagittaria trifolia)-tomato rotation (TY3).

Lines 37-39: "Altogether, compared with the tomato monoculture, the rotations of tomato with eggplant and arrowhead shifted the rhizosphere bacterial communities and improved the yield and quality of the vegetables." Does the increase in yield refer to all three species?

Experimental design

It would be very interesting to insert some pictures of the crops. Is it possible?

Line 128: …" Among them, arrowhead was grown in moist culture." What do you mean by moist culture? In hydroponics? Also, it would be helpful to have pictures of the growths.

Soil sampling

From the description of the sampling, it seems to me that the rhizospheric soil was considered but not the roots. Haven't you considered evaluating endophytic bacteria as well?

RESULTS

Line 244: "3.3. Distribution curves of soil microbial abundance among different planting modes". Perhaps the paragraph number should be removed?

Fig. 4: The legend at the top right in figure 4 is too small and hard to read.

Fig. 6: The graphs e and f (Total soluble sugar) appear to be a duplicate of the same data. Is that so?

And also, I suggest better detailing the caption

Final comment:

Even if English is good, I suggest a language revision to fix some problems.

6. PLOS authors have the option to publish the peer review history of their article (what does this mean?). If published, this will include your full peer review and any attached files.

Reviewer #1: No

Reviewer #2: **Yes: **Elisabetta Franchi

---

## [Author Response · Author response to Decision Letter 0]

11 Nov 2022

Dear Editors and Reviewers,

Thank you for your letter and the reviewers’ comments concerning our manuscript entitled “Rotations improve the diversity of rhizosphere soil bacterial communities, enzyme activities and tomato yield” (ID: PONE-D-22-17572). These comments are all valuable and greatly helpful to revise and improve our paper. Based on the comments, we carefully made modifications. The changes are highlighted in red in the text. We gave responses to the reviewers’ comments one by one as follows:

Reviewer #1: 

1. Comment: The paper is interesting but there are few points that should be revised before publication. The authors should indicate which statistic test they have used to analyze the results (indicate p value also). In general the paper is written in a good English, but more references could be added to support the text. In particular, references in English would add value to the paper. There are ten references in Chinese, please find and include other references in English to support your sentences.

Response: Thanks for your suggestions. The statistical tests used to analyze the results were added and the corresponding p values were provided. (Lines 198-199, 281-282, 294-295, and 309-310)

Several references in Chinese were removed, and 13 English references were added. (Lines 493-510, 544-546, 556-561, 566-568, 572-574, 578-581, 592-595, 618-626) 

2. Comment: In the discussion section, you should not repeat the results without further discuss them. I think you should revise the discussion section trying not to repeat only the results (for example, have a look at lines 311-315)

Response: Thanks for your comments. We rewrote the Discussion referred to other researchers ’studies. The additions are marked in red in the paper. (Lines 319-321, 323, 327-335, 337, 341-348, 357-359, 363-368.)

3. Comment: Keywords: I suggest to use keywords that are not included in the article title. In this way it is probable that more people are reading your paper.

Response: We revised the keywords as suggested. (Line 42)

4. Comment: Line 31: “Lysobacterium” is rarely used in the literature. In order to avoid misunderstandings I suggest to replace it with the more used and known “Lysobacter”.

Response: We replaced “Lysobacterium” with “Lysobacter”. (Line 31)

5. Comment: Line 74: please add references about tomato-rice and tomato-xerophyte crop rotation.

Response: We added 3 references based on the reviewer’s comments. (Lines 493-502)

6. Comment: Lines 75-77: add references to support this sentence.

Response: 2 references were added. (Lines 503-510)

7. Comment: Lines 158-159 Specify the concentration of the sodium phenol solution and of sodium hypochlorite solution used.

Response: We specified the concentrations of the sodium phenol and sodium hypochlorite solutions. (Line 164)

8. Comment: Figure 4: if possible, please insert a bigger legend to the figure. The current legend is too small and difficult to read.

Response: A bigger legend has been added to Figure 4. (Line 261)

9. Comment: Lines 258-259: “TY3 had the highest level of activity.” What are you referring to in this sentence? Are you referring to catalase or sucrase? Please specify.

Response: We are sorry for this confusion. It meant that TY3 had the highest level of catalase activity. It was changed in the text (Line 270)

10. Comment: Lines 260-261: “[…] indicating that the arrowhead-tomato rotation could increase soil fertility.” I think that this sentence should be part of the discussion session. Moreover, you should include more data supporting this sentence to be able to arrive at this conclusion.

Response: This sentence has been moved to the Discussion. This conclusion was obtained based on the an analysis of Figure 5. We rewrote that part in the Discussion. (Lines 369-371, 373-375)

11. Comment: Lines 267-269: These sentences should be part of the discussion section.

Response: We moved these sentences to Discussion. (Lines 357-359, 366-368)

12. Comment: Table 2: Please specify what are the numbers in the table and if they have a unit of measurement. In addition, I suggest to represent the table graphically. This would help the reader to understand quickly the results. Please, specify also the statistic test you used to analyze the data.

Response: We made this correction based on the reviewer’s comments. The numbers in the table and their units of measurement have been noted. In addition, the table was represented graphically along with the statistical test that was used to analyze the data. (Lines 278-284)

13. Comment: Figure 5 and Figure 6: Please specify which statistic test you use to analyze the data.

Response: We made this correction based on the reviewer’s comments. The statistic test was added in the Figure 5 and Figure 6. (Lines 279-282, 292-295)

14. Comment: Figure 6: What is the difference between (e) and (f) ? They are both “Total soluble sugar content”. 

Response: We are sorry for this mistake. Figure 6(f) has been modified. (Line 305)

15. Comment: Line 325: replace “benefit” with “beneficial”.

Response: “benefit” was replaced with “beneficial”. (Line 349)

16. Comment: Lines 341 - 345: Can you find references to support these sentences that are not in Chinese language?

Response: 1 English references to support these sentences were added. (Lines 603-605)

17. Comment: Lines 356 – 357: “Nitrate is an important quality indicator of vegetables.” Please include references to support this sentence.

Response: 1 references were added to support this sentence. (Lines 618-620)

18. Comment: Lines 361 -363: This sentence assumes that sucrase is an indicator of soil fertility. Please include references that support your assumption (possibly in English).

Response: References 40 and 57 were added to provide evidence for the sentence“assumes that sucrase is an indicator of soil fertility.”Both references are in English. (Lines 547-549, 603-605)

19. Comment: Lines 367 -368: I do not understand how this sentence correlates with the rest of the text. Please rephrase and add a reference in English.

Response: The sentence was rephrased and the 62 reference in English was added (Lines 397-398, 621-623)

We wish to profusely thank you for your helpful comments. 

Reviewer #2: 

1. Comment: ABSTRACT

Lines 24-26: …" tomato-tomato continuous cropping (TY1), eggplant-tomato (S. melongena) rotation (TY2) and arrowhead-tomato (Sagittaria trifolia-Solanum lycopersicum) rotation (TY3)." Please standardize the descriptions. For example: tomato-tomato (Solanum lycopersicum) continuous cropping (TY1), eggplant (Solanum melongena) -tomato rotation (TY2) and arrowhead (Sagittaria trifolia)-tomato rotation (TY3).

Response: We made this correction based on the reviewer’s comments. (Lines 24-26)

2. Comment: Lines 37-39: "Altogether, compared with the tomato monoculture, the rotations of tomato with eggplant and arrowhead shifted the rhizosphere bacterial communities and improved the yield and quality of the vegetables." Does the increase in yield refer to all three species.

Response: Yes. Compared to the tomato monoculture, both the rotations of tomato with eggplant and arrowhead can improve the yield of the tomato. (Line 39)

3. Comment: Experimental design

It would be very interesting to insert some pictures of the crops. Is it possible?

Response: It is a good suggestion. Unfortunately, we did not save photographs from the study.

4. Comment: Line 128: …" Among them, arrowhead was grown in moist culture." What do you mean by moist culture? In hydroponics? Also, it would be helpful to have pictures of the growths.

Response: Moist culture does not indicate hydroponics. It means that the soil water content was maintained at 90% when the plants were cultivated, and no water is visible on the surface. Please find the picture below. 

5. Comment: Soil sampling

From the description of the sampling, it seems to me that the rhizospheric soil was considered but not the roots. Haven't you considered evaluating endophytic bacteria as well?

Response: This is a good suggestion, but for this study, we just considered the rhizosphere soil. Our next study will examine the endophytic bacteria in depth.

6. Comment: RESULTS

Line 244: "3.3. Distribution curves of soil microbial abundance among different planting modes". Perhaps the paragraph number should be removed?

Response: The paragraph number was removed. (Line 255)

7. Comment: Fig. 4: The legend at the top right in figure 4 is too small and hard to read.

Response: The figure 4 was changed with a clear legend. (Line 261)

8. Comment: Fig. 6: The graphs e and f (Total soluble sugar) appear to be a duplicate of the same data. Is that so? And also, I suggest better detailing the caption.

Response: Figure 6(f) has been modified, and we provided more detail to the caption based on the reviewer’s comments. (Line 305)

9. Comment: Final comment:

Even if English is good, I suggest a language revision to fix some problems.

Response: We made these corrections based on the reviewer’s comments.

We wish to profusely thank you for your helpful comments. 

We tried our best to improve the manuscript and made changes in the manuscript. These changes will not influence the content and framework of the paper. In addition, we did not list the changes here and instead marked them in red in the revised paper.

We earnestly appreciate the thorough work of the editor and reviewers and hope that the corrections will meet with approval.

Once again, thank you very much for your comments and suggestions.

Best regards!

Yours sincerely,

Cui Feng, Xiaosan Jiang

E-mail: 20142202@jaas.ac.cn, gis@njau.edu.cn (XJ)

---

## [Decision Letter · Decision Letter 1]

26 Dec 2022

Rotations improve the diversity of rhizosphere soil bacterial communities, enzyme activities and tomato yield

PONE-D-22-17572R1

Dear Dr. Jiang,

We’re pleased to inform you that your manuscript has been judged scientifically suitable for publication and will be formally accepted for publication once it meets all outstanding technical requirements.

Kind regards,

Raffaella Balestrini

Academic Editor

PLOS ONE

Additional Editor Comments (optional):

Reviewers' comments:

Reviewer's Responses to Questions

**Comments to the Author**

1. If the authors have adequately addressed your comments raised in a previous round of review and you feel that this manuscript is now acceptable for publication, you may indicate that here to bypass the “Comments to the Author” section, enter your conflict of interest statement in the “Confidential to Editor” section, and submit your "Accept" recommendation.

Reviewer #1: All comments have been addressed

2. Is the manuscript technically sound, and do the data support the conclusions?

Reviewer #1: Yes

3. Has the statistical analysis been performed appropriately and rigorously? 

Reviewer #1: Yes

4. Have the authors made all data underlying the findings in their manuscript fully available?

Reviewer #1: Yes

5. Is the manuscript presented in an intelligible fashion and written in standard English?

Reviewer #1: Yes

6. Review Comments to the Author

Reviewer #1: (No Response)

7. PLOS authors have the option to publish the peer review history of their article (what does this mean?). If published, this will include your full peer review and any attached files.

Reviewer #1: No

---

## [Editor Report · Acceptance letter]

3 Jan 2023

PONE-D-22-17572R1 

Rotations improve the diversity of rhizosphere soil bacterial communities, enzyme activities and tomato yield 

Dear Dr. Jiang:

I'm pleased to inform you that your manuscript has been deemed suitable for publication in PLOS ONE. Congratulations! Your manuscript is now with our production department. 

Kind regards, 

on behalf of

Dr Raffaella Balestrini 

Academic Editor

PLOS ONE